# Extraction of Kidney Anatomy based on a 3D U-ResNet with Overlap-Tile Strategy

Jannes Adam[1], Niklas Agethen[1], Robert Bohnsack[1], René Finzel[1], Timo Günnemann[1], Lena Philipp[1], Marcel Plutat[1], Markus Rink[1], Tingting Xue[1], Felix Thielke[2], and Hans Meine[2,1]

[1] University of Bremen
[2] Fraunhofer MEVIS

**Abstract.** In this paper we present our approach for the KiTS21 Challenge. The goal is to automatically segment kidneys, (renal) tumors and (renal) cysts based on 3D computed tomography (CT) images of the abdomen. The challenge provided public training 300 cases for this purpose. To solve this problem, we used a 3D U-ResNet with pre- and postprocessing and data augmentation. The preprocessing includes the overlap-tile strategy by preparing the input patches, while a rule-based postprocessing was applied to remove false-positive artefacts.

**Keywords:** U-ResNet · residual connection · medical image segmentation

## 1 Introduction

Kidney cancer is a common type of cancer for which automated anatomical labeling would benefit diagnosis and treatment. In clinical practice, however, manual delineation of all relevant structures is too much effort. Therefore, the KiTS challenges have been set up to investigate automated procedures for the extraction of kidney anatomy. The KiTS21 challenge follows the previous KiTS19 challenge which already provided a dataset of labeled kidneys in CT [1]. In addition, KiTS21 provides labels for tumors and cysts. The baseline approach to this problem is a U-Net, of which the nnU-Net variation won the KiTS19 challenge. We try to improve on this baseline with a 3D U-ResNet, data augmentation, pre- and postprocessing. The residual connections of the U-ResNet should reduce the vanishing gradient problem and has also been effectively used in other medical applications [5]. In contrast to the nnU-Net, the overlap-tile strategy is used as it was originally used in the U-Net [4].

This contribution is from the "DeepAnatomy" team of master's students in computer science at the University of Bremen in collaboration with the Fraunhofer Institute for Digital Medicine MEVIS.

## 2    Methods

We trained ca. 40 variations of the U-ResNet to find better parameters and examine the effect of hyperparameters and preprocessing options. In the post-processing we do a rule based cleanup.

### 2.1    Training and Validation Data

Our submission made use of the official KiTS21 training set alone. We converted the annotations in majority voted labels and transversal orientation. From the 300 cases we made a randomized split of 210 training, 30 validation and 60 test cases.

### 2.2    Preprocessing

First the CT values are thresholded at $-1000$ HU. We then resample the voxel size to 1.5mm, which we found best performing for a 3 level network. Adding more levels did not improve the performance, but due of hardware limitations we couldn't train a 5 level network.

The data then gets augmented by x-axis flipping, which swaps the kidney positions, scaling by $\pm 10\%$ and rotation. The rotation is sampled from a normal distribution with a standard deviation of $15°$. We don't use weighted inputs, but the training batches are created with a certain ratio of foreground in it. By doing this, we prevent the model from unlearning rare structures. Analyzing the KiTS21 data showed that cyst voxels are very rare. The batches are generated with a composition of patches, where 50% contain at least one voxel tumor or cyst, 25% at least one voxel cyst and 25% without constraints. Because of the patch size, the average number of cyst voxels in a batch is still smaller than 0.5%.

We separate the dataset images into smaller patches of size 32x32x32 voxels, since the full 3D images do not fit onto our available hardware. The patch size of 32 voxels was chosen together with the batchsize, filter sizes, levels, etc. (described in chapter 2.3) in order to make use of all available GPU memory and have a compromise between statistically independent samples and low overhead.

We use the overlap-tile strategy for seamless segmentation, so the architecture uses valid-mode convolutions and the input images are padded with input image context before being fed into our model to achieve an output patchsize of 32x32x32. The padding size depends on the model architecture and its number of convolution and pooling layers. Our architecture (see chapter 2.3) requires a padding of 21 voxels on both edges of every dimension, resulting in model input images of size 74x74x74 voxels. The padding is implemented by cutting out 74x74x74 patches from the input images (filled-up with $-1000$ HU outside the domain of the original CT volume) so that the output patches of size 32x32x32 voxels are exactly adjacent without overlapping or gaps.

### 2.3   Proposed Method

To meet the challenge we decided to use a 3D U-ResNet architecture, i.e. a U-Net [4] extended by residual blocks. A residual block consists of two 3D convolution layers with kernel size of 3x3x3 and strides of 1x1x1, each followed by Batch Normalization and ReLu activation function (see figure 1).

We setup an U-ResNet with 3 levels, where every level combines a residual block, a dropout layer (with dropout rate 0.2) and another convolution layer with strides 2x2x2 in the down-path (down scaling to reduce the image size) or a transposed convolution layer with strides 2x2x2 in the up-path (up scaling to increase the image size). At each level the number of filters for every convolution layer of that level is doubled, while the first level starts with filters of 32. All the convolution layers of the network apply valid padding, except the layers for up- and down scaling where same padding is used. Aligned with the U-Net implementation the levels from down- and up-path of the same rank are connected via shortcuts.

The first level starts with an additional convolution layer, combined with following Batch Normalization and ReLu layer (down-path), and ends with a convolution with kernel size 1x1x1 that has 4 output channels, one per output class background, kidney, tumor or cyst, followed by Softmax as final activation function (up-path).

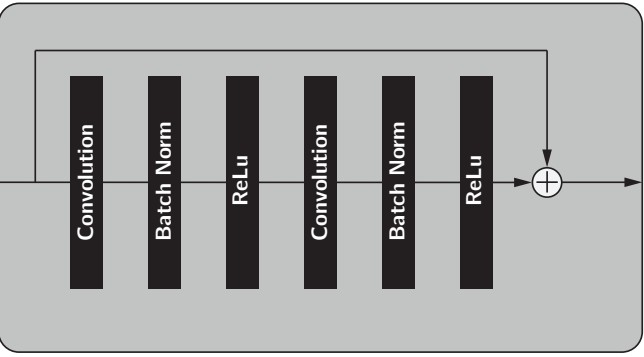

**Fig. 1.** Residual Block

During training batches of 15 patches are fed into the model and a dice loss function is used. As described in chapter 2.2 we apply oversampling instead of class weighting. As optimizer we chose Adam [3] with a learning rate of 0.0001. The remaining parameters are the default ones of the tensorflow parametrization.

The validation during training is done after 800 training iterations based on 500 patches from validation data set, separated into validation batches of 50 patches.

For our validation strategy to find the best model, we built an inference network and evaluated our models using the given evaluation measures of the

challenge. This includes the Sørens-Dice Coefficient and Surface Dice for the three HEC's Kidney and Masses (kidney, tumor, cyst), Kidney mass (tumor and cyst) and tumor. Also, we looked at the Dice and Surface Dice score only for the kidney and cyst alone to see how our scores come up in the HEC's. In the end, we chose the model that had the best Dice and Surface Dice scores for the three HEC's.

## 2.4    Postprocessing

We used connected component analysis to remove false positive fragments not connected to kidneys. This is achieved by retaining only the two components with the largest fractions of voxels labeled "kidney" (1).

Since the lesions were often not classified consistently by our model, so that there are voxels of both classes (tumor and cyst) within a single predicted lesion, an additional postprocessing step homogenizes the classes per lesion. Therefore, a connected component analysis is applied to determine all lesions from our model's output images. For each of these components, we apply a majority vote to adjust the output class of the given component to the class which occurs more frequently (number of voxels) within the given component.

## 3    Results

**Fig. 2.** trainings log

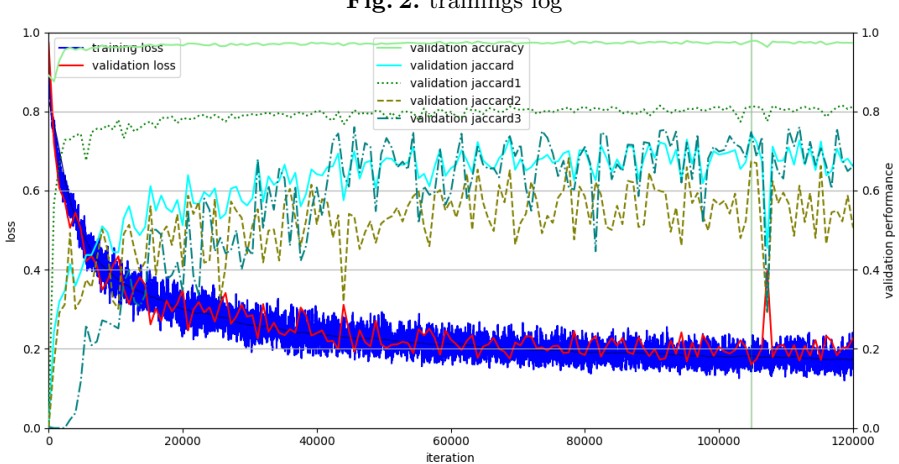

Jaccard 1 to 3 correspond to the classes kidney, tumor and cyst, respectively.

The training took 169601 iterations. Fig. 2 shows the training progression.

In addition to the required metrics, which were measured on our test set, we also considered the Dice of kidney and cysts to better localize sources of

**Table 1.** Scores

| | Dice | | | Surface Dice | | |
|---|---|---|---|---|---|---|
| | Kidney & Masses | Kidney Mass | Tumor | Kidney & Masses | Kidney Mass | Tumor |
| | 0.951 | 0.798 | 0.781 | 0.904 | 0.648 | 0.627 |

| Dice | | Surface Dice | |
|---|---|---|---|
| Kidney | Cyst | Kidney | Cyst |
| 0.841 | 0.360 | 0.838 | 0.210 |

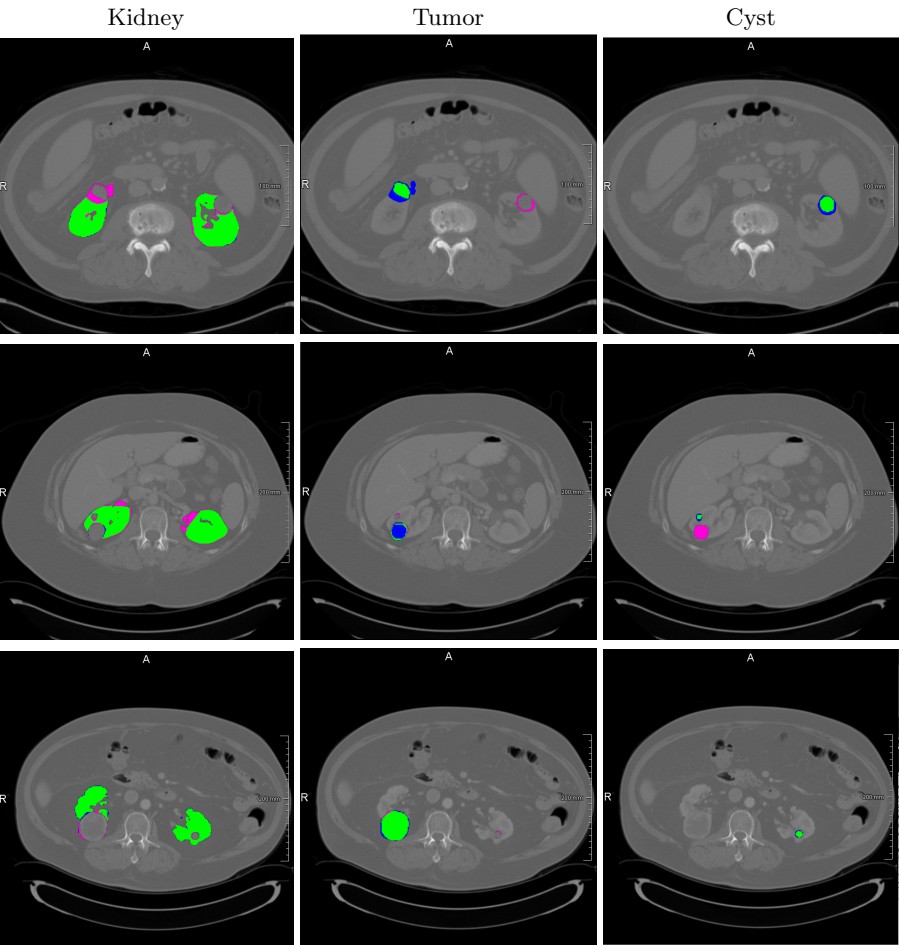

**Fig. 3.** Example segmentation results on three test cases, compared with the reference labels. *green:* true positive, *pink:* false positive, *blue:* false negative

error (see table 1). As can be seen in fig. 3, there are cases where the marginal

regions of the cysts are incorrectly segmented as tumors or cysts and tumors are interchanged. The kidneys, on the other hand, are generally well recognized.

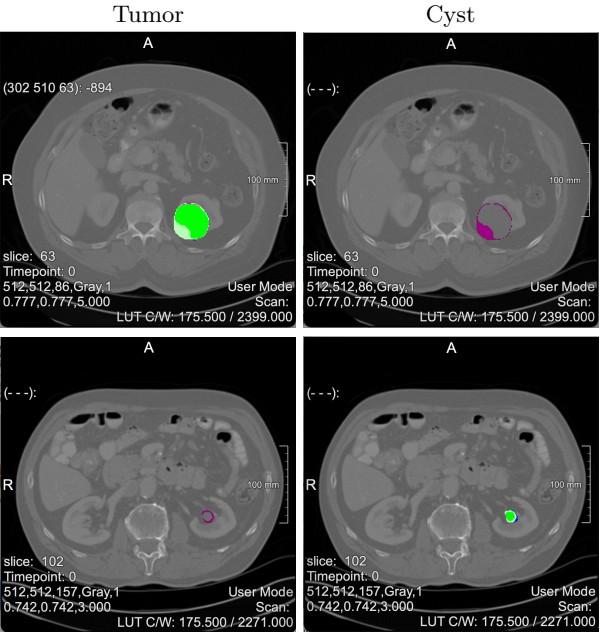

**Fig. 4.** Example segmentation results on two test cases, left: without post-processing, right: with post-processing. *dark green:* true positive both, *light green:* true positive with postprocessing, *dark pink:* false positive without postprocessing

The second step of post-processing corrects the false-positive classified tumor or cyst margins, as can be seen in fig. 4.

## 4    Discussion and Conclusion

While the segmentation of tumors was the most challenging part on the KiTS19 data, segmentation improved significantly with the KiTS21 data using the same model. For this task, identifying cysts and distinguishing between cysts and tumors proved most challenging.

The biggest improvement in cyst segmentation was achieved by oversampling in the batches and adding data augmentation to the pipeline. Additionally, changing the resampling had a great impact on the general performance. This could have been used to chose different voxel spacing for the different structures. One approach could be to start with higher voxel spacing for kidney segmentation and to use this model as a starting point for transfer learning with a lower voxel spacing to segment and distinguish cysts and tumors. We did not do this

as a voxel spacing of 1.5mm proved to be a good compromise between detailed and contextual information.

Regarding the patching we first started with maximum large patches and a batch size of 2 like suggested in the nnU-Net approach [2] to exploit GPUs memory, we then tried different configurations of these two values and noticed that increasing patch size while lowering batch size and vice versa did not impact the performance significantly. We then increased the batch size again to 15 to get a more robust sampling and adjusted the patch size accordingly.

Moreover, there were many false positive structures in the results, which were probably caused by the small voxel size. These were removed by our postprocessing, but this strategy could be extended to address the problem of small cysts on tumor structures and vice versa. One idea would be to check if the volume for a cyst is above a specific threshold, but there were also very small cysts included in the data set. This raises the question of the medical relevance of these very small cysts. In general, we can say that the close study of the data was the key to improvement.

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
