# OpenReview forum: "Extraction of Kidney Anatomy based on a 3D U-ResNet with Overlap-Tile Strategy"
_MICCAI.org/2021/Challenge/KiTS — Submitted to KiTS21 Challenge_

### Official Review · Reviewer_XSUP · 2021-08-30

**Rating:** 9

**Review:**

This paper does an excellent job summarizing this team's challenge submission. As far as I can tell, all template prompts were addressed and figures and tables were employed effectively to support the authors' statements. My only comment is to request that the authors add their final test set results once they are known.

---

### Official Review · Reviewer_RNJn · 2021-08-30

**Rating:** 9

**Review:**

### Overall

- Frontmatter looks good

### Introduction

- Looks good

### Methods

- Please include units when you say you resampled the voxel size to 1.5

### Results

- Very nice

### Discussion and Conclusion

- Looks great

---

### Decision · Program_Chairs · 2021-08-30

**Decision:**

Minor Revisions

**Comment:**

Please address the reviewer comments and resubmit